# Discovery of ACE Inhibitory Peptides Derived from Green Coffee Using In Silico and In Vitro Methods

**DOI:** 10.3390/foods12183480

**Published:** 2023-09-19

**Authors:** Haopeng Dai, Min He, Guilin Hu, Zhongrong Li, Abdulbaset Al-Romaima, Zhouwei Wu, Xiaocui Liu, Minghua Qiu

**Affiliations:** 1State Key Laboratory of Phytochemistry and Plant Resources in West China, Kunming Institute of Botany, Chinese Academy of Sciences, Kunming 650201, China; daihaopeng@mail.kib.ac.cn (H.D.); hemin@mail.kib.ac.cn (M.H.); huguilin@mail.kib.ac.cn (G.H.); lizhongrong@mail.kib.ac.cn (Z.L.); ruma62016@gmail.com (A.A.-R.); wuzhouwei@mail.kib.ac.cn (Z.W.);; 2University of Chinese Academy of Sciences, Beijing 100049, China

**Keywords:** coffee protein, peptide, ACE inhibitor, LC-MS/MS, molecular docking

## Abstract

Inhibition of angiotensin-I converting enzyme (ACE) is an important means of treating hypertension since it plays an important regulatory function in the renin-angiotensin system. The aim of this study was to investigate the ACE inhibitory effect of bioactive peptides from green coffee beans using in silico and in vitro methods. Alcalase and thermolysin were employed to hydrolyze protein extract from coffee beans. Bioactive peptides were identified by LC-MS/MS analysis coupled with database searching. The potential bioactivities of peptides were predicted by in silico screening, among which five novel peptides may have ACE inhibitory activity. In vitro assay was carried out to determine the ACE inhibitory degree. Two peptides (IIPNEVY, ITPPVMLPP) were obtained with IC_50_ values of 57.54 and 40.37 μM, respectively. Furthermore, it was found that two inhibitors bound to the receptor protein on similar sites near the S1 active pocket of ACE to form stable enzyme–peptide complexes through molecular docking, and the Lineweaver–Burk plot showed that IIPNEVY was a noncompetitive inhibitor, and ITPPVMLPP was suggested to be a mixed-type inhibitor. Our study demonstrated that two peptides isolated from coffee have potential applications as antihypertensive agents.

## 1. Introduction

Hypertension or high blood pressure is a major risk factor for several diseases, particularly stroke, heart disease, and chronic kidney disease [1]. It is the leading cause of morality globally, affecting more than 1 billion people with the disease, and is currently experiencing a younger decline. The burden of high blood pressure is disproportionately high in low- and middle-income countries, where two-thirds of cases are present.

Angiotensin-I converting enzyme (ACE), a zinc metallopeptidase, plays a vital role in elevating blood pressure converting angiotensin-I into angiotensin-II, a powerful vasoconstrictor [2]. Due to the evident side effects of synthesized ACE inhibitory medicine (captopril, enalapril), natural ACE inhibitors have become a hot topic of research interest. Therefore, bioactive peptides from natural sources such as animal products, plants, marine resources, microorganisms, and their fermentation products become safe and effective ACE inhibitors to treat hypertension. Animal products such as eggs and milk are rich in protein and are important sources of bioactive peptides. Some oligopeptides with ACE inhibitory activity were found in the egg whites of chicken [3] and ostrich eggs [4]. Protein-rich plant-derived peptides such as beans, grains, and seeds have also been extensively studied in recent years [5,6,7]. Moreover, a variety of bioactive peptides obtained from the protein of marine organisms including fish [8], mollusks [9,10], seaweed [11,12], and microbial fermentation products [13,14] have been confirmed to have an ACE inhibitory function.

With the advance of peptidomics, it has been demonstrated that plant polypeptides exhibit different functions and may have significant bioactivities like being anti-microbial, anti-oxidative, anti-inflammatory, and antihypertensive [15]. The inhibitory effect of peptides on ACE depends on the molecular weight of the active peptide and the hydrophobicity of the three amino acid residues at the C-terminal [16]. Experimental results also suggested that the more hydrophobic amino acids the ACE inhibitory peptide contains, the easier it is to enter the hydrophobic region of the enzyme active center, thereby interacting with the amino acid residues to inhibit its activity [17]. The main method for obtaining plant peptides is to hydrolyze proteins with different proteases [18]. Alcalase and thermolysin are two commercialized microbial proteases, which are commonly used to hydrolyze proteins from food sources to prepare active oligopeptides. In the current study, these two enzymes were used due to their characteristics of enzymatic cleavage of hydrophobic amino acids and broad action sites.

Coffee is one of the most valuable traded commercial drinks in the world, especially *Coffea arabica*, which occupied over 55% of the world’s coffee production in 2021 [19]. Long-term research studies have shown that caffeine, chlorogenic acids (CGAs), and trigonelline are the main compounds with diverse bioactivities [20]. In recent years, a variety of novel diterpenes have been demonstrated existing in green Arabica coffee beans in addition to cafestol and kahweol [21,22]. Coffee beans, the “fruits and seeds of plants of the genus *Coffea*” ref. [23], containing about 10% protein [24] which plays an important role in the growing stage, can be considered as a natural source of bioactive peptides. Peptides from green coffee beans have been studied using simulated gastrointestinal digestion through in vitro hydrolysis and in silico prediction [25]. Although both computer predictions and experimental results indicated that green coffee protein hydrolysates may have biological activities, there was limited information about the identification of peptides. Peptides isolated from spent coffee grounds (SCGs) have also been studied for their antioxidant and ACE inhibitory activity using enzymatic hydrolysis [26] and fermentation by bacteria [27]. The Maillard reaction occurs during the roasting step, resulting in denaturation of protein and generation of melanoidins, which might cause interference during extraction and in vitro evaluation of activity [26]. So far, there are few studies in the literature describing the oligopeptides with an ACE inhibitory effect obtained from the protein hydrolysates of green coffee. As a result, it is of great significance to discover bioactive peptides and investigate their bioactivity and inhibition mechanism.

This research was to explore active peptides with an ACE inhibitory effect in green coffee beans using the method of peptidomics. Identification, screening, and characterization of these active peptides were conducted, and molecular docking was also used to explore the kinetics and mechanism of inhibition.

## 2. Materials and Methods

### 2.1. Chemicals and Samples

The green coffee beans of *coffea arabica* were purchased from Baoshan, Yunnan Province of China.

Thermolysin, dithiothreitol (DTT) were purchased from Mreda Technology Inc. Alcalase, sodium dodecyl sulfate (SDS) were purchased from Shanghai Macklin Biochemical Co., Ltd., (Shanghai, China). Tris(hydroxymethyl)aminomethane (Tris) was purchased from Shanghai Acmec Biochemical Co., Ltd. Acetonitrile (ACN) for LC-MS/MS was produced by oceanpak alexative chemical, Ltd. Angiotensin converting enzyme (ACE), captopril, 2-[4-(2-hydroxyethyl)piperazin-1-yl]ethanesulfonic acid (HEPES), N-[3-(2-furylacryloyl)]-L-phenyalanyl-glycyl-glycine (FAPGG), trifluoroacetic acid (TFA), and formic acid (FA) were purchased from Sigma-Aldrich (Shanghai, China). Sodium chloride, potassium phosphate dibasic, and potassium phosphate monobasic were purchased from Sangon Biotech Co., Ltd. (Shanghai, China). Hydrochloric acid and sodium hydroxide were supplied by Damao Chemical Reagent (Tianjin, China). Distilled water was obtained from a Milli-Q system from Millipore.

### 2.2. Protein Extraction

The protein extraction procedure was similar to that described by Pérez-Míguez et al. [28] with some minor modifications; 25 g of grounded green coffee beans were defatted four times with 500 mL petroleum ether; 1 g of flour was extracted with 100 mL of 100 mM Tris-HCl buffer (pH 7.5) containing 1% SDS and 0.25% DTT under ultrasonic for 15 min at room temperature. The mixture was centrifuged at 4000 r/min at room temperature for 15 min and supernatants were collected. Protein in the supernatant was precipitated with 100 mL of cold acetone at 4 °C for 12 h and collected by centrifugation for 20 min at room temperature. The precipitate was dialyzed with distilled water overnight followed by lyophilization and stored at −20 °C.

### 2.3. Protein Digestions

Protein extracts were dissolved in the 5 mM phosphate buffer (pH 8.0) at the concentration of 2.5 mg/mL and submitted to enzymatic hydrolysis with alcalase (10,000 U/g substrate) and thermolysin (0.05 g enzyme/g substrate) separately at 50 °C for 2 h with slight stirring, according to the proposed procedure of Esteve et al. [29] The digestions were stopped by increasing the temperature to 90 °C for 20 min, and supernatants were collected by centrifugation at 10,000 g and 4 °C for 10 min.

### 2.4. LC-MS/MS Identification of Peptides

Solid phase extraction (SPE) was carried out using a C18 SPE Cartridge purchased from MREDA for removal of salts prior to nano-LC-MS/MS analysis: (i) conditioning with 2 × 3 mL 100% acetonitrile (ACN); (ii) washing with 3 × 1.5 mL 0.1% trifluoroacetic acid (TFA); (iii) loading of sample (3 mL, containing 0.1~0.2% TFA); (iv) washing with 3 × 1.5 mL 0.1% TFA; (v) collecting peptides after elution with 1.5 mL 50% ACN followed by 1.5 mL 80% ACN [30].

Identification of the desalted sample was then conducted by LC-MS/MS using a Triple TOF 6600^+^ mass spectrometer in positive ion mode with a OptiFlow Turbo V ion source, coupled to a NanoLC 400 (Ekspert); 8 μL of hydrolysate was loaded on a trap Nano LC precolumn at 10 μL/min for 3 min. Thereafter, the sample was loaded at 5 μL/min onto a Micro LC Column (3 μm, ChromXP C18, 120 Å, 0.3 × 150 mm) at the column temperature of 40 °C for the separation of peptides. Mobile phase A was 2% ACN containing 0.1% formic acid (*v*/*v*), and mobile phase B was 98% ACN containing 0.1% formic acid (*v*/*v*). Elution gradient employed for separation was: 3–22% B in 40 min, 22–35% B in 10 min, 35–80% B in 1 min, 80% B for 3 min, 80–3% B in 1 min, and 3% B for 5 min. The following settings were applied: nebulizer (gas 1) 25 psi, heater (gas 2) 50 psi, curtain gas 35 psi, and ionization temperature 150 °C. TOF-MS scan range was set at 350 to 1500 *m*/*z*, followed by an MS/MS scan at 100 to 1500 *m*/*z*.

The raw data obtained from the LC-MS/MS was processed using the software MaxQuant (version 2.3.0.0) [31,32] (https://www.maxquant.org/ (accessed on 26 February 2023)), and the protein database of coffee (*Coffea arabica*) was searched from NCBI.

### 2.5. In Silico Prediction and Analysis of Bioactive Peptides

The bioactivities of all identified peptides were evaluated using in silico methods. The ACE inhibitory capacity of peptides was predicted using the AHTpin platform (https://webs.iiitd.edu.in/raghava/ahtpin/ (accessed on 1 March 2023)) in accordance with the support vector machine (SVM) classification model [33]. Peptides with an SVM score greater than 0 were regarded as antihypertensive peptide inhibitors. In addition, the physiochemical properties (hydrophobicity and isoelectric point) of peptides were analyzed by the PepDraw server (http://pepdraw.com/ (accessed on 28 February 2023)).

### 2.6. Peptide Synthesis and ACE Inhibitory Activity

The peptides predicted with potential antihypertensive capacity were synthesized by GenScript. The ACE inhibitory capacity of hydrolysates and selected peptides were measured as previously described with minor modifications [34]; 400 μL of samples (hydrolysates, synthesized peptides dissolved in the HEPES-NaCl buffer at a concentration of 1.25 mg/mL, 125 μM separately) were mixed with 100 μL of ACE (0.2 U/mL). After incubation of the mixture at 37 °C for 10 min, 500 μL of FAPGG was added and the absorbance value at 340 nm was detected instantly by a FlexStation 3 and recorded as A_0 min_. After being kept at 37 °C for 30 min, the absorbance value at 340 nm was detected again and recorded:$$ACE\;inhibitory\;rate\left(\%\right)=\frac{{∆A}_{control}-{∆A}_{sample}}{{∆A}_{control}}\times 100\%$$
where A_0 min_ is the initial absorbance at 340 nm; A_30 min_ is the absorbance at 340 nm after keeping at 37 °C for 30 min, and ΔA is the reduction in the absorbance: A_0 min_–A_30 min_.

Referring to the methods in the literature, and combined with the above in vitro activity assay results, the design of the inhibition pattern test is as follows: various concentrations (0, 0.2, 0.4 mM) of ACE inhibitory peptides were added to each reaction system; meanwhile, the concentration of FAPGG were set as 0.25, 0.5, 1, 2 mM. According to the Michaelis–Menten kinetic equation, the Michaelis–Menten constant (K_m_) and the maximum reaction rate (V_max_) were calculated from the Lineweaver–Burk plot to determine the inhibition pattern of peptides. All evaluations were performed in triplicate.

### 2.7. Gastrointestinal Digestion Simulation Study

In vitro gastrointestinal digestion simulation was performed according to the method described by Fan et al. [3] with minor modifications. Two peptides (CP2, CP3) were dissolved in ultrapure water at 1.0 mg/mL, respectively. The solutions were adjusted to pH 2.0 with 0.1 HCl. Then, pepsin (1% E/S, *w*/*w*) was added to peptide solutions and the mixtures were incubated at 37 °C for 2 h. Half of the two digests were terminated in digestion by increasing the temperature to 90 °C. After the other half were adjusted to pH 7.5 with 0.1 NaOH, pancreatin (1% E/S, *w*/*w*) was added and the systems were digested at 37 °C for 2 h. The reactions were then stopped by boiling for 10 min. Subsequently, solvents were centrifuged at 10,000 rpm for 10 min. The supernatants were lyophilized to collect and analyzed using Agilent 6545 series Q-TOF mass spectrometer (Agilent, Waldbronn, Germany). The ACE inhibitory activity of the digested peptide solutions was determined under the same conditions.

### 2.8. Molecular Docking

Molecular docking was conducted using Autodock Vina software (version 1.5.7) to further study the mechanism of interaction between ACE (receptor) and peptides (ligands) according to methods previously reported with slight modification [35]. The crystal structure of ACE was obtained from the RSCB protein data bank (https://www.rcsb.org/ (accessed on 13 March 2023)) with the PDB code of 1O86. The structures of selected peptides were generated by ChemDraw. The number of points in the X-, Y-, and Z-dimensions were set as 75, 75, and 75, and spacing was set as 1Å. PyMol software (version 2.5.4) was used to visualize the pose of ligand from the docked complex.

## 3. Results

### 3.1. Peptide Identification

The hydrolysates were identified with MaxQuant software (version 2.3.0.0) for each mass spectral data generated by LC-MS/MS. The results showed that 55 and 22 peptides were identified from alcalase and thermolysin hydrolysates, respectively. The identified peptides ranged in length from 7 to 15 amino acids and had molecular masses ranging from 600 to 1400 *m*/*z*. Appendix A summarized the peptides after digestion, along with their experimental data. Figure 1 shows, as examples, the MS/MS spectra obtained for two of the identified peptides: GLPSGGAPSGY (Gly-Leu-Pro-Ser-Gly-Gly-Ala-Pro-Ser-Gly-Tyr) in the alcalase hydrolysate and ITPPVMLPP (Ile-Thr-Pro-Pro-Val-Met-Leu-Pro-Pro) in the thermolysin hydrolysate. It is worth noting that no common peptide fragments were found in the coffee protein hydrolysate after enzymatic hydrolysis by two proteases.

It has been reported that alcalase has a wide range of applications in food production and processing due to broad selectivity and specificity of the cleavage site of alcalase [36]. The results have shown that among the products hydrolyzed by alcalase, the C-terminal amino acid residues of peptides were mostly Gln, Tyr, Met, Lys, and Leu. In addition, it was also found that in a certain proportion of peptide fragments, the hydrophobic amino acid was close to the N-terminal, which is very common in bioactive peptides [37]. In contrast, thermolysin hydrolysate presented a smaller number of peptides, only 22 peptides were identified, and mostly produced large fragments under this hydrolysis condition. The results in Appendix A showed that a great number of peptides have Leu, Phe, Ile, and Val at the N-terminal position, which was consistent with literature [38], and the presence of large bulky amino acids in these peptide sequences suggested potential antihypertensive capacity.

### 3.2. In Silico Screening

Peptides with a score greater than 60 indicating the degree of matching between the secondary mass spectrum and their databases, were submitted to analyze hydrophobicity and isoelectric point and predict potential bioactivity using the in silico method. The physiochemical properties analyzed by PepDraw server and ACE inhibition prediction score of peptides hydrolyzed by alcalase are shown in Table 1, and those of peptides hydrolyzed by thermolysin are shown in Table 2.

The AHTpin platform was used to screen antihypertensive peptides in accordance with the SVM classification models. With the existence of the AHTpin platform, the computational prediction of potential antihypertensive peptides from the hundreds of thousands of peptide sequences can be performed in a short time. The 71 identified peptides were entered to predict probability in ACE inhibition; 16, 15 peptides were authenticated as antihypertensive peptides because their SVM scores were higher than 0 in alcalase, thermolysin hydrolysates, respectively [33]. The higher the score, the higher the ACE inhibitory activity of peptides; when the SVM score is > 1, the peptide may have a relatively higher potential with ACE inhibitory activity [39]. According to the result, five coffee bean peptide sequences with an SVM score of > 1.0 were chosen. The five selected sequences GLPSGGAPSGY, IIPNEVY, ITPPVMLPP, VLETPDGPL, and VKNPNPIPIP were named CP1, CP2, CP3, CP4, and CP5.

### 3.3. In Vitro Evaluation of ACE Inhibitory Capacity

The selected monomer peptides were synthesized by the solid-phase synthesis method with a purity of ≥95%. The ACE inhibitory capacity of five synthesized peptides predicted by AHTpin with potential antihypertensive activity were determined. The inhibition rate (%) of two enzyme hydrolysates and five peptides separately are shown in Figure 2.

Thermolysin was the enzyme yielding peptides with the higher ACE inhibitory capacity, and the hydrolysate of alcalase also had considerable ACE inhibitory activity, according to results. Generally, the antihypertensive peptides contained between 2 and 12 amino acids, and presented a certain amount of hydrophobic residues [40,41]. Compared with captopril, the ACE inhibition rate of thermolysin hydrolysate reached 73.3%, indicating that there are very promising inhibitors among them, and these inhibitors are completely safe without any side effects.

The ACE inhibitory capacity of CP 1–5 was tested at a concentration of 50 μM. The results showed that five peptides exhibited various degrees of inhibitory capacity from 6.0 to 53.3%. The inhibition rates of the two peptides of CP2 (IIPNEVY) and CP3 (ITPPVMLPP) were significantly higher than those of the other three peptides, which were 49.3% and 53.3%, respectively. Their IC_50_ values were 57.54 ± 7.78 μM and 40.37 ± 5.56 μM, respectively. Among them, the inhibition rate of CP4 (VLETPDGPL) was the lowest, only 6.0%. Five peptides obtained by computational screening all showed certain activities, suggesting that the prediction results have reference value.

### 3.4. Inhibition Kinetics of Bioactive Peptides

The inhibition models of ACE inhibitory peptides mainly include competitive inhibitors, noncompetitive inhibitors, and mixed-type inhibitors and their inhibition mechanisms are different. To determine the inhibitory kinetic pattern of the ACE inhibitory peptides obtained from green coffee protein, Lineweaver–Burk plots were constructed, and the results are shown in Figure 3. As the concentration of IIPNEVY (CP2) increased, the Michaelis–Menten constant (K_m_) remained unchanged, while the V_max_ decreased, which is characteristic of the noncompetitive inhibition pattern indicating that this peptide bound to essential sites outside the active part of enzyme. The reported noncompetitive inhibitors include VFDGVLRPGQ from rice bran protein [34], QLDL from the mycelia of *Ganoderma Lucidum* [42], and VGLPPNSR and QAGLSPVR from tilapia skin gelatin [43], etc. As for the peptide ITPPVMLPP (CP3), Figure 3B showed that the fitted curve did not intersect the x-axis at the same point and the K_m_ values of control and various inhibitory peptide concentration were not the same, suggesting that the inhibition of CP3 at different concentrations presented a mixed-type inhibition pattern. The V_max_ values decreased as the peptide concentration increased and the K_m_ increased with the addition of an inhibitory peptide. The result was similar to the ACEI peptides YGIKVGYAIP from palm kernel cake protein hydrolysates [44].

### 3.5. Molecular Docking Studies

To further explain the inhibitory mechanism between the identified peptides and ACE, the docking study between CP2 or CP3 and ACE was performed. Prior research have suggested that the binding pockets related to the activity of ACE are mainly distributed in S1(Ala354, Tyr523, and Glu384), S2(Gln281, Lys511, His353, Tyr520, and His513), and S1’(Glu162) [45]. However, the docking results showed that CP2 or CP3 did not form any interactions with amino acid residues of ACE active pockets. As shown in Figure 4, two peptides bound to the receptor protein in the same cavity at the upper end of ACE close to active pocket S1 and tended to combine with the similar amino acid residues. Both peptides produced an H-bond very close to the critical residue Ala354 in an ACE active pocket. Moreover, CP3 bound with Arg522 in pocket S1 exhibiting a higher inhibition rate according to the in vitro assay. We hypothesized that the ability of CP2 was to generate interactions with nonactive sites, resulting in a distortion of the catalytic configuration, which is consistent with the inhibition kinetic pattern of bioactive peptides as ACE noncompetitive inhibitors. When CP3 bound to the receptor protein, it formed more complex interactions with the amino acid residues near the active sites (Ser355, Arg522), suggesting it may be a potent mixed-type inhibitor.

### 3.6. In Vitro Gastrointestinal Digestion Stability

As ACE inhibitory peptides exert physiological regulatory functions after oral administration, there is a risk of being degraded by proteases in the gastrointestinal tract. Therefore, the stability of the two inhibitory peptides against gastrointestinal digestion was analyzed using LC-MS characterization and variations in the ACE inhibition rate. The MS results of CP2 after pepsin hydrolysis were consistent with that of control (Figure 5A,B). After successive hydrolysis by pepsin and pancreatin, new peptide sequences were detected (Figure 5C). Compared with before digestion, the ACE inhibition rate of CP2 decreased after being digested by pepsin, and then after digestion by pancreatin, its inhibition rate decreased significantly (Table 3). The result was consistent with the LC-MS analysis, indicating that the CP2 is well resistant in gastric digestion, but may be affected by pancreatin. Among the products from in vitro digestion (pepsin, pepsin-pancreatin) of CP3, new peptides were determined. Only a small amount of CP3 was degraded in the digestion by pepsin (Figure 5E), and the result of the ACE inhibition assay also suggested that CP3 was partially degraded during pepsin hydrolysis as the inhibition rate did not change significantly. Great changes in the MS analysis and ACE inhibitory activity occurred after incubation with pepsin-pancreatin. Thus, pancreatin may be hydrolyzing the CP3 and its pepsin digestion product into smaller peptide fragments with a higher ACE inhibitory effect. The result of this simulation has suggested that CP3 may be a pro-drug peptide which could be converted to true inhibitors by gastrointestinal proteases, resulting in increased activity [46].

## 4. Conclusions

In the current study, traditional methods coupled with emergent peptidomics methods were employed to discover potential antihypertensive peptides in green coffee. Five peptides derived from green coffee that may have ACE inhibitory activity were predicted through in silico screening. However, subsequent in vitro activity tests showed that the experimental results were slightly different from the prediction results. The hydrolysates obtained when using proteases both showed considerable ACE inhibitory capacity. Combined with the identification of peptides, it was considered that thermolysin and alcalase cleave peptide bonds near hydrophobic amino acid residues, resulting in peptides containing favorable amino acid residues for antihypertensive activity at the C-terminal. Based on the analysis of the above results, the peptide with the highest experimental inhibition rate was not the one with the highest score, indicating that the result predicted by the webserver AHTpin may be inaccurate. Additionally, some peptides with equivalent or stronger activities were missing, leading to limitations in finding active peptides effectively. In the screening results, two novel oligopeptides, IIPNEVY and ITPPVMLPP, exhibited excellent ACE inhibitory activity with IC_50_ values of 57.54 and 40.37 μM. Lineweaver–Burk plots showed that the former peptide was a noncompetitive inhibitor and the latter might act in a mixed-mode type of inhibition. It was found that CP2 bound to the nonactive sites in ACE through molecular docking study, thereby destroying the natural conformation of ACE and affecting its activity. CP3 bound to the receptor protein in the same cavity to cause inhibition of ACE. However, future research should focus on active peptides that may be missed because of the high inhibition rate of coffee protein hydrolysate, and other potential peptides in coffee hydrolysate are worthy of further exploration. Moreover, in vitro gastrointestinal digestion simulation has been conducted to investigate biostability, and the CP3 can be considered to be a so-called pro-drug peptide, while the CP2 requires further experiments to explore its delivery method. In summary, peptides derived from the hydrolysates of Coffea arabica are a promising source for the treatment of high blood pressure.

## Figures and Tables

**Figure 1 foods-12-03480-f001:**
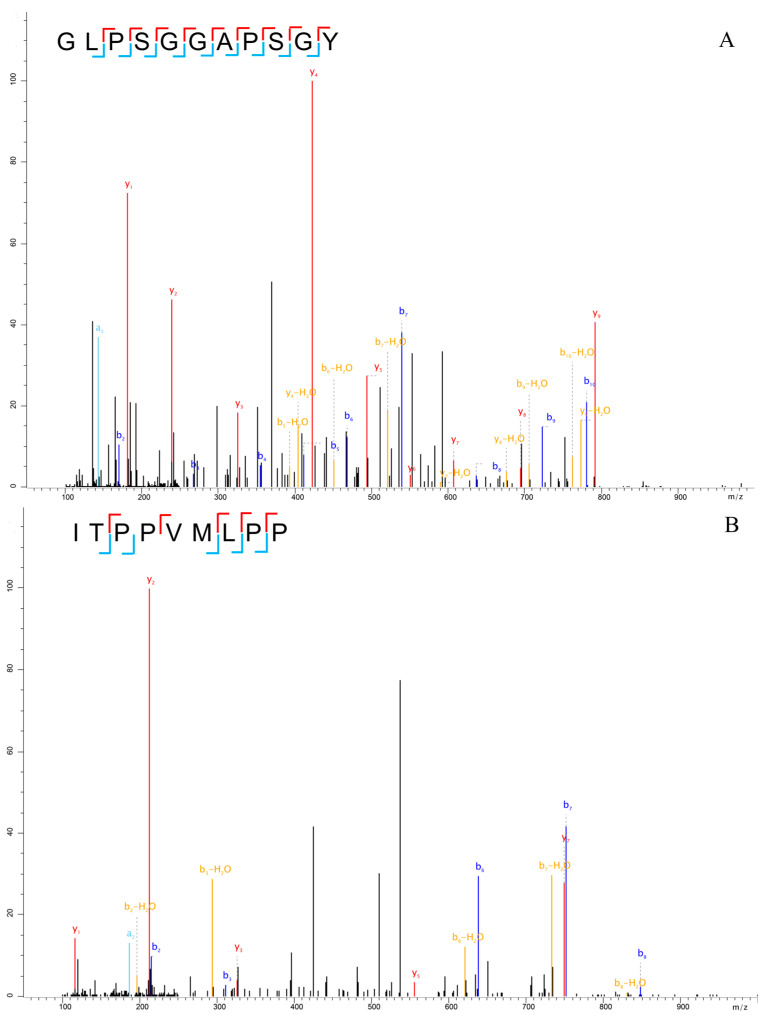
MS/MS spectra of identified peptides GLPSGGAPSGY (**A**), ITPPVMLPP (**B**).

**Figure 2 foods-12-03480-f002:**
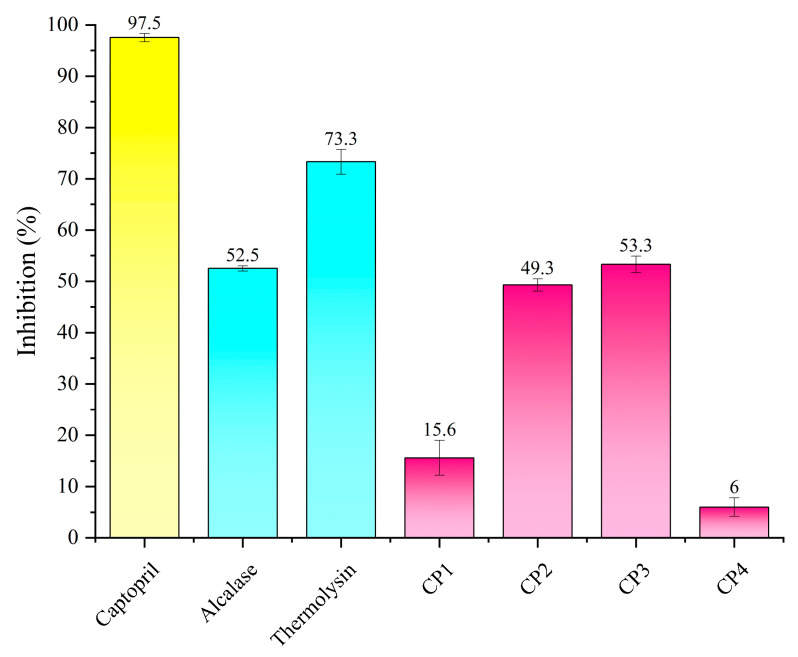
ACE inhibitory capacities of the green coffee bean protein isolate digested with alcalase, thermolysin at the concentration of 500 μg/mL, five screened peptides identified from protein hydrolysates at the concentration of 50 μM, and captopril at the concentration of 0.1 μM.

**Figure 3 foods-12-03480-f003:**
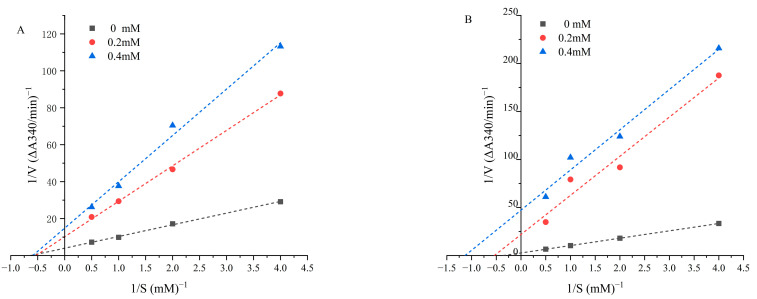
Lineweaver–Burk plots of CP2 IIPNEVY (**A**) and CP3 ITPPVMLPP (**B**) for ACE inhibition. ACE inhibitory activities by different concentration (0, 0.2, 0.4 mM) of peptides were measured at various substrate concentrations (0.25, 0.5, 1.0, 2.0 mM).

**Figure 4 foods-12-03480-f004:**
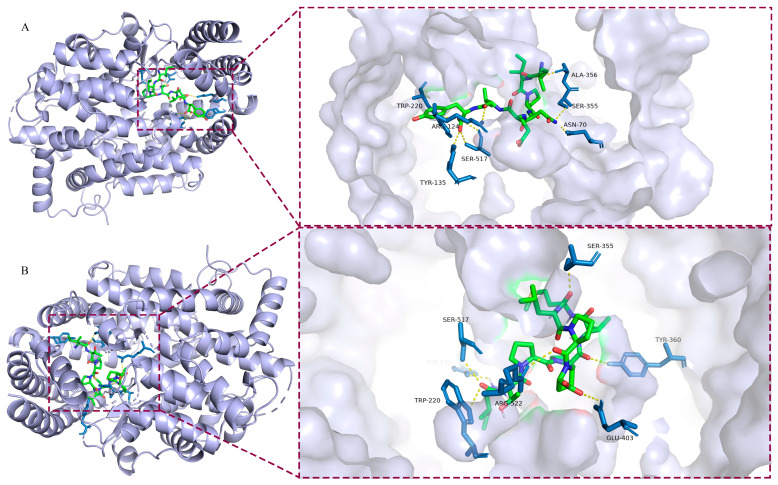
Molecular docking of bioactive peptides binding on receptor protein: (**A**) 3D illustration of the interaction between CP2 and ACE; (**B**) 3D illustration of the interaction between CP3 and ACE.

**Figure 5 foods-12-03480-f005:**
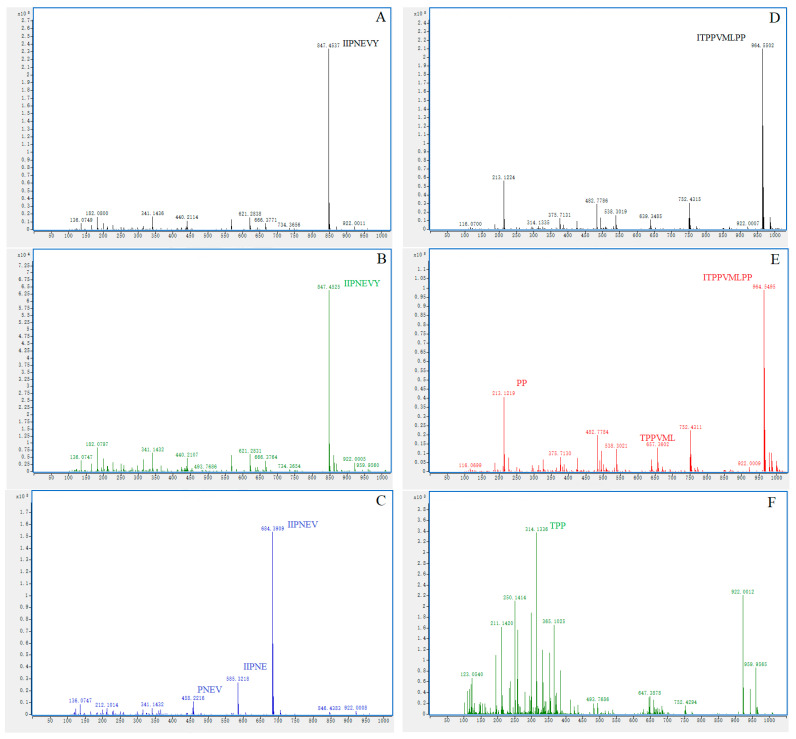
(**A**) Mass spectrum of CP2; (**B**) Mass spectrum of CP2 digested by pepsin; (**C**) Mass spectrum of CP2 digested by pepsin-pancreatin; (**D**) Mass spectrum of CP3; (**E**) Mass spectrum of CP3 digested by pepsin; (**F**) Mass spectrum of CP3 digested by pepsin-pancreatin.

**Table 1 foods-12-03480-t001:** Physiochemical properties and SVM score of identified peptides from the alcalase hydrolysates.

No.	Peptide Sequence	Score	Hydrophobicity (kcal/mol)	Isoelectric Point	SVM Score
1.	SENIGLPQ	189.5	12.53	3.12	0.00
2.	ILLPGFTQ	187.6	4.88	5.44	−0.75
3.	VVINPGNPTGQ	186.4	11.16	5.44	−0.38
4.	GLPSGGAPSGY	185.1	12.24	5.45	1.47
5.	TNDNAMINPL	166.5	11.94	3.12	−0.94
6.	FEDNAGVIVNPK	157.7	17.71	4.00	−0.05
7.	SPVAPLAPVTL	155.9	6.61	5.50	0.18
8.	GESFWGGQ	152.4	12.41	3.12	0.58
9.	EGDGGVGTIKL	150.6	19.99	4.01	−1.57
10.	GIESVPAALIGL	138.8	10.23	3.20	−1.14
11.	RAIPEEVL	134.7	14.78	4.09	0.07
12.	SAERGFLY	133.6	11.78	6.58	−0.43
13.	AFNVDLK	129.4	12.27	6.76	−1.15
14.	GLPASPGAAVGQ	128.4	12.65	5.44	0.80
15.	VADPDKLPTIPGQ	126.0	18.24	3.92	0.47
16.	NLGSIPTQ	125.8	9.15	5.25	−0.43
17.	ADLSRIDL	125.5	14.33	3.93	−0.46
18.	SAFRAIPE	124.1	12.11	6.61	−1.15
19.	ALDPGLTY	122.2	10.37	3.05	0.49
20.	IQIIFPE	120.9	7.37	3.09	−0.53
21.	APIAVGDVIPDGTL	120.9	14.60	2.94	−1.23
22.	GQLIIVPQ	119.3	6.78	5.44	−1.04
23.	ILMIGTQ	117.6	5.91	5.44	−1.80
24.	GVKSVEIL	116.5	12.65	6.83	−0.27
25.	TNEILIGK	114.5	13.09	6.55	−1.30
26.	APILDEVAVSL	113.5	12.23	2.98	−0.31
27.	ALRALPE	111.3	11.98	6.98	−0.41
28.	ADSLDLRL	107.4	14.20	3.93	0.10
29.	AKDPVRVL	105.1	14.62	10.20	0.20
30.	KNPNIPDPNTL	105.0	15.19	6.44	0.72
31.	SDVGLERQ	103.8	17.65	4.00	−1.13
32.	AGPGGWNDPDML	103.8	16.25	2.94	0.12
33.	TVDKRLL	103.0	13.44	9.82	−0.75
34.	KNPNIPDPNTLM	102.1	14.52	6.44	0.17
35.	SALRAIPE	101.5	12.57	6.61	−1.11
36.	GYIPGIIY	100.9	5.56	5.43	−0.53
37.	RVDSIPIL	100.1	10.00	6.42	−1.47
38.	GDAPRVL	98.4	13.43	6.76	−0.27
39.	VASGNVL	98.0	8.69	5.58	−0.93
40.	VIEGDLL	97.6	12.24	2.98	−1.55
41.	ALATPLL	96.3	5.54	5.59	−0.30
42.	AITPPVMLPPL	94.7	4.46	5.59	0.51
43.	LILGPDSPAVQ	93.3	10.62	3.04	0.60
44.	IPLDLNY	71.4	8.20	3.05	0.31
45.	DIIEFIQ	70.0	10.87	2.91	−1.40
46.	IIPNEVY	67.0	9.11	3.14	1.04
47.	GGKADVL	67.0	15.43	6.73	−0.79
48.	VGHTDTARMLL	63.6	14.20	7.89	−1.26
49.	METSNSVPSIL	62.0	10.65	3.20	−0.54
50.	AATLPLM	61.3	6.12	5.41	−1.60

Light, dark shades correspond to peptides with SVM score higher than 0.0, 1.0, respectively.

**Table 2 foods-12-03480-t002:** Physiochemical properties and SVM score of identified peptides from the thermolysin hydrolysates.

No.	Peptide Sequence	Score	Hydrophobicity (kcal/mol)	Isoelectric Point	SVM Score
1.	LITMEPNSL	176.0	8.94	3.20	−0.80
2.	IFDPFPSD	164.0	11.38	2.78	0.33
3.	ITPPVMLPP	163.3	5.21	5.23	1.33
4.	VLETPDGPL	148.2	13.89	2.98	1.42
5.	FVDPDGWKT	141.5	15.26	3.92	0.29
6.	FWDSNNPE	141.0	13.67	2.87	−0.74
7.	FLPEYSEQ	128.6	12.86	2.96	−0.26
8.	LFPSPSPPPP	128.6	6.70	5.23	0.21
9.	IGLPQEAD	128.2	15.36	2.82	0.51
10.	AVNHPNFPST	117.9	11.25	7.95	−0.34
11.	VMKNRPISEE	107.0	18.97	6.98	0.28
12.	VKNPNPIPIP	104.4	10.26	10.14	2.31
13.	VVGDPLDPNSHHGPQ	99.8	22.47	4.98	0.86
14.	LLERGPTPEP	98.2	16.29	4.08	0.75
15.	IDWKETPEAHV	90.1	21.15	4.32	0.35
16.	YSPDGEEGFPGNL	88.1	20.17	2.90	0.97
17.	FHPPGSDRVD	81.3	19.04	5.14	−0.27
18.	VMDDTSESKPQHPSR	80.5	27.30	5.27	0.38
19.	IDWKETPEAH	78.9	21.61	4.31	0.43
20.	FDDEVKQGQL	72.6	20.88	3.69	−1.29
21.	FRFPSEAG	67.0	12.17	6.65	0.28

Light, dark shades correspond to peptides with SVM score higher than 0.0, 1.0, respectively.

**Table 3 foods-12-03480-t003:** ACE inhibition rate of the peptides following digestion by gastrointestinal proteases.

Enzyme	ACE Inhibition Rate (%)	
	CP2 (IIPNEVY)	CP3 (ITPPVMLPP)
Control ^a^	49.3	53.3
Pepsin	36.6	54.9
Pepsin-Pancreatin	9.8	69.6

^a^ Peptides without any treatment.

## Data Availability

The data used to support the findings of this study can be made available by the corresponding author upon request.

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
