# Peer review of "Discovery of ACE Inhibitory Peptides Derived from Green Coffee Using In Silico and In Vitro Methods"

_foods, 2023, doi:10.3390/foods12183480_

Round 1
Reviewer 1 Report
Dear authors,
The manuscript entitled Discovery of ACE Inhibitory Peptides Derived from Coffee using In silico and In vitro method is well written. However, I do not feel that the works are of strong enough novelty or impact to be published to Q1 journals. If authors further add the content to the work, i.e., in vitro cell culture and its molecular study of to report that green coffee bean (Coffea arabica) acted as ACE inhibitors and , then I believe it would be sufficient. Thank you.
Author Response
Dear Reviewers:
Thank you for your letter and for the reviewers' comments concerning our manuscript entitled "Discovery of ACE Inhibitory Peptides Derived from Coffee Using in silico and in vitro Methods" (ID: foods-2565959). Those comments are all valuable and very helpful for revising and improving our paper, as well as the important guiding significance to our researches. We have studied comments carefully and have made correction which we hope meet with approval. Revised portion are marked in red in the paper. The main corrections in the paper and the responds to the reviewer's comments are as flowing:
In the current study, we discovered two novel ACE inhibitory peptides in green coffee protein hydrolysates. Furthermore, using computer screening, molecular docking and kinetic modeling to explore bioactive peptides from food sources, there were few literature reports on the study of green coffee bean bioactive peptides.
Thank you for your suggestion on molecular study, this is exactly what we need to work on in the future, we are also working on this aspect, and you will see this research in our future work.
Reviewer 2 Report
Discovery of ACE inhibitory peptides derived from coffee using in silico and in vitro methods were the subject of research in this study. This research was to explore active peptides with ACE inhibitory effect in green coffee beans using method of peptidomics. Identification, screening, and characterization of these active peptides were conducted, molecular docking was also used to explore the kinetics and mechanism of inhibition. However, the authors did not clearly point out what was new in their study.
In the main title, I suggest inserting "green coffee" instead of "coffee", and that "in silico" and "in vitro" be written in italics as well.
The topic in this study is original, relevant and specific in this field of research.
However, the authors should emphasize this in more detail, in the manuscript itself. For example, such as: originality, relevance, specificity, novelty..., of your research topics in this area.
Why did they choose "green coffee"? Somehow, it is explained very briefly and not convincingly in the Introduction.
The manuscript is interesting and deals with a current and innovative topic in the field of human health.
However, he would have a suggestion to improve the manuscript in the Introduction to include more information about peptides with similar function from e.g. other sources (I recently had the opportunity to review a manuscript where the source was nuts), which would further emphasize the importance of the topic of this manuscript in relation to similar previously published materials.
The design of the experiment is not completely clear.
It should be explained why certain trials were chosen. As various concentrations (0.2, 0.4 mM) of ACE inhibitory peptides were added to each 136 reaction system; various concentration of FAPGG were set as 0.25, 0.5, 1, 2 mM, various substrate concentrations (0.25, 0.5, 1.0, 2.0 mM). How were all these concentrations chosen?
What is "control"? It appears for the first time in Figure 3.
This should be explained and defined earlier (say in the experiment plan).
In Figure 1, it is not clear what the upper and lower images represent.
It should be marked, for example with A, B, and an explanation given in the name of Figure 1.
What is "control"? It appears for the first time in Figure 3.
The current conclusion is general and short. In line with the previous comments, the conclusion should be improved.
The listed references are appropriate. It is possible that they will have to be supplemented, due to previously given comments.
Author Response
Thank you for your letter and for the reviewers' comments concerning our manuscript entitled "Discovery of ACE Inhibitory Peptides Derived from Coffee Using in silico and in vitro Methods" (ID: foods-2565959). Those comments are all valuable and very helpful for revising and improving our paper, as well as the important guiding significance to our researches. We have studied comments carefully and have made correction which we hope meet with approval. Revised portion are marked in red in the paper. The main corrections in the paper and the responds to the reviewer's comments are as flowing:
Point 1: In the main title, I suggest inserting "green coffee" instead of "coffee", and that "in silico" and "in vitro" be written in italics as well.
Response 1: "coffee" has been replaced by "green coffee" in the manuscript. "in silico" and"in vitro" has been rewriten in italics in the manuscript.
Point 2: the authors should emphasize this in more detail, in the manuscript itself. For example, such as: originality, relevance, specificity, novelty..., of your research topics in this area.
Response 2: We have carefully revised the content based on this suggestion, please review the revised manuscript.
Point 3: Why did they choose "green coffee"? Somehow, it is explained very briefly and not convincingly in the Introduction.
Response 3: We have made revision in the Introduction section. We believed that compared with roasted coffee, the Maillard reaction does not occur in green coffee, which will not cause protein denaturation and generation of melanoids to interfere with subsequent experiment. And there were still few studies in the literature describing the bioactive peptides obtained from the protein hydrolysates of green coffee.
Point 4: he would have a suggestion to improve the manuscript in the Introduction to include more information about peptides with similar function from e.g. other sources (I recently had the opportunity to review a manuscript where the source was nuts), which would further emphasize the importance of the topic of this manuscript in relation to similar previously published materials.
Reponse 4: We have carefully enriched the Introduction section based on this suggestion. Please review revised manuscript.
Point 5: It should be explained why certain trials were chosen. As various concentrations (0.2, 0.4 mM) of ACE inhibitory peptides were added to each 136 reaction system; various concentration of FAPGG were set as 0.25, 0.5, 1, 2 mM, various substrate concentrations (0.25, 0.5, 1.0, 2.0 mM). How were all these concentrations chosen?
Response 5: By referring to the experimental methods of the previous research literatures, in order to make the obtained data reflect the conclusion intuitively and clearly, we chose these substrate (FAPGG) concentrations through multiple ratios. The inhibitor (ACE inhibitory peptides) concentrations were chosen by combining the in vitro activity assay results when inhibitor concentration was 50μM and references.
Point 6: What is "control"? It appears for the first time in Figure 3.
Response 6: “control” refers to the inhibitor concentration is 0mM. It has been replaced by "0mM" in the Figure 3, showing more clearly.
Point 7: It should be marked, for example with A, B, and an explanation given in the name of Figure 1.
Response 7: It has been marked with "A" and "B" in the Figure 1
Point 8: The current conclusion is general and short. In line with the previous comments, the conclusion should be improved.
Response 8: It has been enriched in the manuscript to make it more readable. Please review the revised manuscript.
Point 9: The listed references are appropriate. It is possible that they will have to be supplemented, due to previously given comments.
Response 9: It has been supplemented as suggested
Reviewer 3 Report
The manuscript reports the identification of peptides with ACE-inhibiting activity. The experimental plan is well structured and sound.
Presentation can be improved, especially of the tables. I would suggest a 3D-type of presentation of all the data.
It would be intersting also to convert the tables into plots with the score on the X in decreasing order, and the other features as lines. Numbering peptides entries as 1-25 or 1-50, depending on the table. Peptide sequences can then be presented in the figure caption.
Enrich the discussion mentioning other reported activities of the most significant peptides identified, e.g. the TOP10 have other known activities?
The language is very good,.
Author Response
Thank you for your letter and for the reviewers' comments concerning our manuscript entitled "Discovery of ACE Inhibitory Peptides Derived from Coffee Using in silico and in vitro Methods" (ID: foods-2565959). Those comments are all valuable and very helpful for revising and improving our paper, as well as the important guiding significance to our researches. We have studied comments carefully and have made correction which we hope meet with approval. Revised portion are marked in red in the paper. The main corrections in the paper and the responds to the reviewer's comments are as flowing:
Point 1: Presentation can be improved, especially of the tables. I would suggest a 3D-type of presentation of all the data.
Response 1: We tried to modify the presentation of the content in the table to make the results more vivid, but the content (identification score, hydrophobicity, isoelectric point, and SVM score) in the table is not relatively relevant, and it will be messy displayed in the plot and affect the reading, so we considered retaining the table to display the results.
Point 2: Enrich the discussion mentioning other reported activities of the most significant peptides identified, e.g. the TOP10 have other known activities?
Response 2: Discussion section has been enriched based on suggestion. Some other reported bioactivite peptides have been mentioned in other sections
Reviewer 4 Report
Authors describe a study in which they ifnd ACE inhibitory peptides in coffe beans subjected to enzymatic digestion with alcalase or thermolysin. They identified produced peptides analyzing extracts by LC-MS/MS using a Q-TOF instrument. Then they performed in silico and in vitro experiments on most promising ACE inhibitory peptides to study the molecular docking and the percentage of inhibition of ACE enzyme. The study seem to be well conducted however, in my opinion, in its present form it is not suitable for publication as no discussion of the results achieved is present.
I suggest authors to include a discussion section in which they (i) explain why alcalase and thermolysin were choosen as enzymes to treat coffee beans, (ii) discuss their findings in comparison with other articles dealing with ACE inhibitory peptides from food, (iii) discuss how their findings can be reasonably useful to treat human hypertension. Additionally, a part of the results can be moved into the new discussion section (e.g. lines 209-212, lines 221-224, lines 257-262).
Following here are listed some comments to improve the quality of the article:
Line 17: in my opinion in this study there was no de novo sequencing, just a database search, otherwise in materials and methods section this must be explained in details.
Line 35: recently, ACE inhibitory peptides have been found also in protein extracts from yeast, insects (Tenebrio molitor), Muscovy Duck (Cairina moschata) and spirulina (Arthrospira platensis), not only in plants. For sake of completeness, it is worth mentioning some additional studies in the introduction or discussion section given that they describe alternative sources of ACE inhibitory peptides derived from food or food supplements (DOI: 10.1021/acs.jafc.0c06053; 10.3390/nu14163288; 10.3390/foods12010050; 10.1016/j.peptides.2019.170107).
Line 189: please briefly describe in this section why you choose SVM score > 1 as cut-off value to select potential ACE inhibitory peptides
Line 190: “higher potential to be used as ACE inhibitors” with respect to …..? Captopril? To the other candidate peptides?
Table 1. peptide n. 11 showed a SVM score > 0 but it is not highlighted, please modify
Section 3.3: did you conducted such experiments also in presence of a competitor? Have you performed the same experiments using Captopril in order to compare the % of inhibition obtained for the 5 potential inhibitory peptides with respect to it?
Did you take in consideration that such peptides may be further cleaved by gastro-intestinal enzymes such as trypsin, pepsin or chymotrypsin when orally administered? Have you tried to simulate in silico digestion with these enzymes? Some considerations about this fact can be also included in the discussion section before suggesting that “peptides derived from hydrolysates of Coffee Arabica are promising source for the treatment of high blood pressure”.
English is fine. Few sentences need to be modified (e.g. lines 186-188: ...only 16 and 15 peptides...; or lines 119-120: ...were considered as ACE inhibitors.)
Author Response
Thank you for your letter and for the reviewers' comments concerning our manuscript entitled "Discovery of ACE Inhibitory Peptides Derived from Coffee Using in silico and in vitro Methods" (ID: foods-2565959). Those comments are all valuable and very helpful for revising and improving our paper, as well as the important guiding significance to our researches. We have studied comments carefully and have made correction which we hope meet with approval. Revised portion are marked in red in the paper. The main corrections in the paper and the responds to the reviewer's comments are as flowing:
Point 1: in my opinion in this study there was no de novo sequencing, just a database search, otherwise in materials and methods section this must be explained in details.
Response 1: After discussion, we modified "de novo" to "database searching" as suggested
Point 2:recently, ACE inhibitory peptides have been found also in protein extracts from yeast, insects (Tenebrio molitor), Muscovy Duck (Cairina moschata) and spirulina (Arthrospira platensis), not only in plants. For sake of completeness, it is worth mentioning some additional studies in the introduction or discussion section given that they describe alternative sources of ACE inhibitory peptides derived from food or food supplements (DOI: 10.1021/acs.jafc.0c06053; 10.3390/nu14163288; 10.3390/foods12010050; 10.1016/j.peptides.2019.170107).
Response 2: We have mentioned some additional studies in the introduction to increase the integrity of the content.
Point 3: please briefly describe in this section why you choose SVM score > 1 as cut-off value to select potential ACE inhibitory peptides
Response 3: It has been mentioned in 2.5. We have also re-writen the description in 3.2 highlighted in red.
Point 4: “higher potential to be used as ACE inhibitors” with respect to …..? Captopril? To the other candidate peptides?
Response 4: "... they have a relatively higher potential to be used as ACE inhibitors" wtih respect to the other candidate peptides.
Point 5: peptide n. 11 showed a SVM score > 0 but it is not highlighted, please modify
Response 5: peptide No. 11 has been highlighted.
Point 6: did you conducted such experiments also in presence of a competitor? Have you performed the same experiments using Captopril in order to compare the % of inhibition obtained for the 5 potential inhibitory peptides with respect to it?
Response 6: We would conduct such a experiment in the presence of competitor to further explore the mechanism of inhibition, we are working on this aspect. We have performed the experiment using Captopril at a concentration of 0.1μM, and the result was shown in the attachment. We believed that without consideration for its safety, Captopril was a great antihypertensive agent.
Point 7: Did you take in consideration that such peptides may be further cleaved by gastro-intestinal enzymes such as trypsin, pepsin or chymotrypsin when orally administered?
Response 7: Thank you for your suggestion on in vitro simulation of human gastrointestinal digestion study, this is exactly what we need to work on in the future, we are also working on this aspect, and you will see this research in our future work.
Point 8: Have you tried to simulate in silico digestion with these enzymes?
Response 8: We have tried to simulate in silico digestion using BIOPEP-UWM (https://biochemia.uwm.edu.pl/en/start/). However, alcalase hydrolysis cannot be simulated on this website.

Round 2
Reviewer 1 Report
Dear authors, thank you for your reply, however, I am still hold to my previous comment. Good luck.
Previous comment:
The manuscript entitled Discovery of ACE Inhibitory Peptides Derived from Coffee using In silico and In vitro method is well written. However, I do not feel that the works are enough to be published to Q1 journals. If authors further add the content to the work, i.e., in vitro cell culture and its molecular study of to report that green coffee bean (Coffea arabica) acted as ACE inhibitors and , then I believe it would be sufficient. Thank you
English is fine.
Author Response
Thank you for your letter and for the reviewers' comments concerning our manuscript entitled "Discovery of ACE Inhibitory Peptides Derived from Green Coffee Using in silico and in vitro Mehtods" (ID:foods-2565959). Those comments are all valuable and very helpful for revising and improving our paper, as well as the important guiding significance to our researches. We have studied comments carefully and have made correction which we hope meet with approval. Revised portion are marked in blue in the paper. The main corrections in the paper and the responds to the reviewer's comments are as flowing:
We further studied the gastrointestinal stability of two novel peptides to make the research more readable. Under the current conditions of our laboratory ,in vitro cell culture study cannot be completed in the short term. We will repenlish the laboratory equipment and conditions as soon as possible to carry out further studies.
Reviewer 2 Report
I would like to thank the authors for their consideration of previously given comments and revision and improvement of the quality of the manuscript.
The authors, taking into account the previously given revision comments, significantly improved their manuscript.
I feel that the manuscript is now significantly improved and can be considered more acceptable for publication.
Author Response
Thank you for your letter and for the reviewers' comments concerning our manuscript entitled "Discovery of ACE Inhibitory Peptides Derived from Green Coffee Using in silico and in vitro Mehtods" (ID:foods-2565959). We appreciate Reviewers,warm work earnestly. Once again, thank you very much for your comments and suggestions.
Reviewer 4 Report
Authors made some text editing, expecially in the introduction section and in colclusion section, but still the manuscript needs improvements. I suggest authors to include a discussion section instead of extending the conclusion section. Moreover I strongly suggest authors to improve and discuss the following points that were only partially taken in consideration during the first revision process:
1. authors did not included a brief explanation about the cut-off value > 1 in SVM. In my opinion it is not enough to cite a previous work. It is important to add one or two sentences explaining why this cut-off was chosen to help readers understanding author's choice. Especially in light of the result reported by authors in the conclusion secion: "However, subsequent in vitro activity tests showed that the experimental results were slightly different from the prediction results" (lines 300-301). By applying 2 selection criteria (i.e. AHTpin platform and SVM), probalby authors lost some potentially interesting peptides.
2. authors are working on additional ACE inhybition experiments in presence of competitors, and comparing the inhibitory potential to captopril. My suggestion was to include results of such experiments in this study, not in future studies. These results are needed to support author's conclusions.
3. authors are planning to simulate gastrointestinal digestion in a future study to investigate the bioavailability of potential ACE inhibitory peptides. Why were not performed additional tests to improve significance and quality of the present manuscript? Even such results are needed to support author's conclusions.
Author Response
Dear reviewers:
Thank you for your letter and for the reviewers' comments concerning our manuscript entitled "Discovery of ACE Inhibitory Peptides Derived from Green Coffee Using in silico and in vitro Mehtods" (ID:foods-2565959). Those comments are all valuable and very helpful for revising and improving our paper, as well as the important guiding significance to our researches. We have studied comments carefully and have made correction which we hope meet with approval. Revised portion are marked in blue in the paper. The main corrections in the paper and the responds to the reviewer's comments are as flowing:
Point 1: authors did not included a brief explanation about the cut-off value > 1 in SVM. In my opinion it is not enough to cite a previous work. It is important to add one or two sentences explaining why this cut-off was chosen to help readers understanding author's choice. Especially in light of the result reported by authors in the conclusion secion: "However, subsequent in vitro activity tests showed that the experimental results were slightly different from the prediction results" (lines 300-301). By applying 2 selection criteria (i.e. AHTpin platform and SVM), probalby authors lost some potentially interesting peptides.
Response 1: We have re-write the explanation in blue. Those are not two selection criteria, in AHTpin, machine learning technique 'support vector machine (SVM)' is used to predict and screen antihypertensive peptides.
Point 2: authors are working on additional ACE inhybition experiments in presence of competitors, and comparing the inhibitory potential to captopril. My suggestion was to include results of such experiments in this study, not in future studies. These results are needed to support author's conclusions.
Response 2: Under the current conditions of our laboratory ,this study cannot be completed in the short term. And we briefly discussed the inhibitory potential compared to captopril
Point 3: authors are planning to simulate gastrointestinal digestion in a future study to investigate the bioavailability of potential ACE inhibitory peptides. Why were not performed additional tests to improve significance and quality of the present manuscript? Even such results are needed to support author's conclusions.
Response 3: In vitro gastrointestinal digestion simulation has been completed and the results were presented in the revised manuscript in blue.